# Accreditation of Quality in Primary Health Care in Chile: Perception of the Teams from Accredited Family Healthcare Centers

**DOI:** 10.3390/ijerph20032477

**Published:** 2023-01-30

**Authors:** Juan Coss-Mandiola, Jairo Vanegas-López, Alejandra Rojas, Pablo Dubó, Maggie Campillay-Campillay, Raúl Carrasco

**Affiliations:** 1Escuela de Obstetricia y Puericultura, Facultad de Ciencias Médicas, Universidad de Santiago de Chile, Santiago 8320096, Chile; 2Departamento de Enfermería, Facultad de Ciencias de la Salud, Universidad de Atacama, Copiapó 7500015, Chile; 3Facultad de Ingeniería y Negocios, Universidad de Las Américas, Santiago 3981000, Chile

**Keywords:** accreditation of quality, primary health care, quality of health care, primary healthcare centers

## Abstract

This study aimed to investigate the perception of the health teams belonging to the Family Healthcare Centers (CESFAMs) that are accredited, regarding the process of implementation and the achievement of accreditation. A qualitative approach was applied, with contributions from grounded theory, through the technique of individual in-depth interviews and focus groups. The interviews were carried out in nine accredited CESFAMs. For the presentation, organization and analysis of the data, Atlas.ti V9 software was used. From the results, derived from the open phase of the analysis, obtained from the opinions of the participants, a total of 26 categories emerged relating to the facilitating and hindering factors of the process. From the axial phase, it was possible to establish central categories that were related to quality management policies, the structure of Primary Health Care (PHC), participation and co-construction, and leadership and change management. In conclusion, the discourse of the teams reveals the need to have necessary conditions for the accreditation process, which are mainly related to training, characteristics of the types of leadership and teamwork in harmony with the process. Finally, the study reveals a gap in the community participation in this process, which suggests continuing this line of research.

## 1. Introduction

The accreditation process is a strategy in the quality of care by which the Health Services increase the probability of obtaining the desired results according to their health policies [1,2].

According to the International Society for Quality in Health Care (ISQua), accreditation is a process of self-assessment and external peer review used by social services and health care organizations to accurately assess their level of performance in relation to established standards and to implement ways to continuously improve the health or social care system [3]. This evaluation of health care is a process that compares predefined and evidence-based requirements or standards, against the use of standardized quantitative standards and qualitative metrics, to evaluate, recommend improvements and report on quality levels from the clinical perspective and, organizationally, in the health facilities or organizations in which it is applied, a program that can take many forms around the world, using different frameworks, approaches, standards and results but sharing the same goal [4,5].

Knowing the impact of accreditation has important implications at the level of public policies and health providers for the restructuring of the accreditation implementation process [6,7].

Many investigators positively correlate accreditation and quality improvements, and therefore patient satisfaction [8,9,10].

A study that aimed to assess the impact of accreditation in public primary and secondary health care institutions in Kerala, India, found that accreditation affects all the structure, process and outcome domains of care delivery. In primary care settings, and these results confirm Donabedian’s model [11], the structural and interpersonal elements of a health care environment impact patient safety [6,7].

Findings from this same research can provide important information for redefining priorities when implementing or planning the implementation of accreditation in Kerala or elsewhere and forms a basis for policy development in health care reforms [6,7].

The literature indicates that culture, as an element to be evaluated in an accreditation process, has a substantial impact on different aspects of performance, while other organizational/structural characteristics have limited evidence of a significant effect on the performance of health teams [12,13].

Therefore, although the organizational/structural characteristics of a PHC center can affect its performance, culture is the main player in this game [14].

That is why, if a PHC center strives to improve the quality and safety of its clinical services, it must evaluate and improve the culture related to patient safety, because the culture of a PHC center is reflected in the decisions made in terms of safety, and this affects the performance of the accreditation of this health facility [14].

In Chile, the Ministry of Health, adhering to international commitments, installs care security as a strategic focus on the quality of care [15]. It is in this sense that accreditation appears, which is regulated by Law 19,937, which in its article 4, number 11, indicates that it is the Ministry of Health, which corresponds, among other functions, to “establish the standards minimums that must be met by institutional health providers, such as hospitals, clinics, doctor’s offices and medical centers, in order to guarantee that the services reach the quality required for the safety of users [16]. These standards will be set according to the type of establishment and the levels of complexity of the benefits and will be the same for the public and private sectors. It must set standards regarding sanitary conditions, safety of facilities and equipment, application of techniques and technologies, compliance with care protocols, competencies of human resources, and in any other matter that affects the safety of benefits”. It also adds, in the same law, that it must “establish an accreditation system for institutional providers authorized to operate” [16].

In Chile, in practical terms, accreditation enters into force from the publication of the regulations for the accreditation system for institutional health providers, according to Supreme Decree No. 15 of 2007 of the Ministry of Health, which establishes in its article that “the accreditation of authorized, public and private institutional health providers will be carried out in accordance with these regulations. Said process will be intended to evaluate compliance, by those institutional providers that submit to it, of the standards set to ensure that the benefits they provide have the necessary quality to protect the safety of their users” [17]. According to the records of the Superintendency of Health, from 2009 onward, the first accredited institutional providers began to be observed in Chile [18]. Although Chile has enacted a health reform since 2005, which promotes a comprehensive health model with a community focus and PHC, in practice, the emphasis placed on this level of primary care has not favored its management processes, including the accreditation of the quality of its health facilities [19]. Proof of this is that, until 2019, only 16 of 273 communes with PHC centers in the country had accredited CESFAMs [20]. This leads to argue that there is inequality in the accreditation process, given that, as of September 2022, this figure has not increased significantly, with only 32 communes having accredited CESFAMs, which is equivalent to a meager 11% of the total communes in the country that have this type of health establishment [18].

Regarding the factors that influence a Family Healthcare Center (CESFAM) to achieve accreditation, a previous study carried out by our research team concluded that variables of a financial nature were not decisive in achieving accreditation for CESFAMs. However, it is possible that there are other factors that may be facilitating the achievement of the accreditation processes. Therefore, adequately knowing how the communes behave with establishments that did manage to comply with said process can lead to improvements in municipal management policies that favor the accreditation process in PHC establishments [19].

A study carried out in public health centers in Kuwait obtained aspects such as administrative support, staff training on accreditation and expansion in the application of electronic systems as facilitating factors for accreditation implementation. On the other hand, study participants reported various challenges in implementing accreditation, including personnel-related challenges (e.g., a heavy workload and burdens imposed by accreditation requirements), organizational resource and system-related challenges (e.g., poor teamwork among health center professionals and inadequate infrastructure in some facilities) and patient-related challenges (e.g., a poor understanding of accreditation) [21].

We know well that there is a disparate appreciation of the acceptance of the accreditation process by health teams. A good example of this is the Catalan Accreditation model, which was well accepted by the pilot teams, with great motivation from the majority of the participants, also promoting the quality of the health system at the first level of health, acting in this way as a guarantor of quality for citizens and, at the same time, providing confidence to professionals [22]. A second example is found in the establishment of the quality model in Andalusia, Spain, which does not generate a consensus among the different interest groups [23]. What is interesting about this second case is that the study was carried out using a qualitative methodology, which guides us to use a similar methodology.

It is necessary to point out that in a quality accreditation process, as in other organizational change processes, there are several interest groups which are likely to state different discourses passively, unconsciously and uncritically, or actively as agents involved, responsible and aware of the constructive implications that adopting one discourse or another may entail. Consistent with this, just as in the Chilean quality system, the Andalusian quality model is not implemented without problems, as a unique and monological discourse, but rather the official discourse “breaks” in its landing in a multiplicity of voices: different versions of the organizational reality [23].

Derived from the above, it is expected to collect and take advantage of this multiplicity of discourses that arise around the process of the implementation and achievement of accreditation in Chile and, in the end, translate them into lessons learned and share these experiences with the teams of centers of non-accredited health, including a qualitative analysis that allows determining certain facilitating or hindering factors to the process of the accreditation of quality in PHC establishments [19].

Therefore, this study aimed to investigate the perception of the health teams belonging to the CESFAMs that are accredited, regarding the process of the implementation and achievement of accreditation.

## 2. Materials and Methods

### 2.1. Study Design

The study considers a qualitative approach using contributions from grounded theory and through data collection techniques based on individual in-depth interviews and focus groups.

### 2.2. Target Group

The target group of the study corresponded to professionals, technicians, administrative and support assistants who belong to CESFAMs that were accredited until March 2020. For the selection of the establishments, the official records of accredited providers were used, available at the Superintendency of Health [18], in this case, classified as Low-Complexity Open Care Providers and with the CESFAM category.

The recruitment of the CESFAMs was performed through a formal letter requesting participation, sent electronically to each accredited establishment, which they then had to return signed and stamped by the highest authority of CESFAM. After sending the mail, an attempt was made to achieve a telephone connection. In the end, due to accessibility and feasibility (commitment) of their participation, the study was completed with 9 establishments out of a total of 42 that were accredited up to that date.

### 2.3. Data Collection

The data collection was conducted with individual and group interview technique. The individual interview was applied to professionals, quality managers, who led the implementation and achievement of accreditation in health establishments. The focus group interviews were applied to two groups: (1) professionals/members of the leading team; (2) health technicians, administrative and support assistants in each health center.

The collection of information was carried out using a detailed guideline that considered the following categories of topics to be dealt with in interviews and focus groups:Perception regarding the culture of evaluation and accreditation of quality in health.Perception about general aspects of the implementation of the quality management policy and achievement of quality accreditation in PHC establishments.Facilitating and hindering factors of the implementation and achievement of quality accreditation in the health establishment in areas such as: administrative, organizational, others.(a)Facilitating factors for the operation of the internal network in the areas administrative, organizational, financial and other aspects that the interviewee wants to comment on.(b)Factors hindering the operation of the internal network in the areas administrative, organizational, financial and other aspects that the interviewee wants to comment on.Assessment and meanings represented by the implementation and achievement of accreditation quality in their health facility.

### 2.4. Sampling Strategy

The sampling strategy was of an opinionated and homogeneous type and sought a discourse focused on one aspect of the topic studied. Therefore, the study participants had similar characteristics. It is a strategy that reduces and simplifies the understanding of the phenomenon.

### 2.5. Verbatims

As planned, 100% of the interviews considered to be key informants of the process were conducted: quality manager, professionals and technical-administrative personnel. The verbatims, according to Amezcua [24], correspond to phrases expressed by the informants and in an exact manner, which fill the proposed categories (codes) with content. They correspond to the evidence of the interpretation made by the main investigators.

### 2.6. Presentation and Analysis of Results

The interviews were recorded, rigorously transcribed and anonymized by assigning a code that allowed the meaning of the content to be associated with each of the participants. The categories (codes) are described below following the structure of the axial phase and with the help of the Atlas Ti^®^ V9 analysis program.

It should be noted that the due authorization of the Institutional Ethics Committee of the University of Santiago de Chile was obtained, which analyzed and authorized all the documents used in this study, the informed consents, the guidelines for individual and group interviews and the letters of commitment of each CESFAM.

## 3. Results

This study included a total of nine CESFAMs, belonging to three regions and four communes of the country. The results describe some main characteristics of the participants, and then the phases that allowed for obtaining the central categories and subcategories of the analysis that allow for summarizing the accreditation experience of the different participating groups are reported.

Regarding the individual interviews, a total of nine professionals in charge of quality participated, of whom three were kinesiologists, three nurses, one midwife, one psychologist and one nursery educator. The average age of all participants was 38.7 years, with a mean of 35 years and an age range of 31–65 years. On average, seniority in the position is 4.7 years (see Table 1).

Regarding the group interviews, these are divided into professional groups and Higher-Level Nursing Technicians (TENS) and administrative groups. The professionals group was made up of 42 members. A total of 33.3% were men and 66.7% women, with an average age of 40.6 years. A total of 11 nurses, 8 midwives, 6 kinesiologists, 4 social workers, 4 pharmaceutical chemists, 4 dentists, 2 doctors, 2 engineers and 1 early childhood educator participated, with an average time in office of 6.9 years. Meanwhile, the TENS and administrative group were made up of 45 members. A total of 26.7% were men and 73.3% were women, with a mean age of 43.8 years. A total of 22 TENS participated, 19 administrative, 2 service assistants, 1 administration engineer and 1 computer scientist, with an average time in their position of 13 years.

### 3.1. Processes to Define the Categories and Subcategories for the Analysis

Two phases were established, the open phase and the axial phase. The open phase allowed the identification of conceptual categories and codes “in vivo”, which identified the main meaning patterns in the stories. Meanwhile, in the axial phase, the categories were associated according to the common properties or characteristics found, with the purpose of reducing the categories and giving greater conceptual strength. The presentation, organization and analysis of the data were carried out with the support of the Atlas.ti version 23 software [25].

### 3.2. Open Phase

Twenty-six categories were identified, which emerge from the different opinions of the members of the health teams of the CESFAMs. These categories are listed below: (1) organizational aspects, (2) training, (3) characteristics of quality team, (4) evaluation culture, (5) participatory management team, (6) accreditation preparation strategies, (7) quality training, (8) role of manager in the quality process, (9) implementation of the quality management policy, (10) interest in management, (11) leadership from people in charge of quality, (12) motivations to accredit, (13) participation of the authority, (14) team participation, (15) perception of when it was implemented, (16) meanings of accreditation achievement, (17) time assigned to quality manager, (18) team rating, (19) financial aspects, (20) organizational aspects, (21) characteristics of the quality team, (22) technical-clinical difficulties, (23) role of manager in the quality process, (24) quality team functions, (25) impact of the COVID-19 pandemic and (26) infrastructure for the quality process.

### 3.3. Axial Phase

In this phase, the categories were determined by looking for common properties or characteristics that allowed their association. At this point, the central categories and subcategories were established, which made it possible to more precisely describe the experience of the health teams involved in the quality and accreditation processes (see Table 2).

The results are described below according to the central analysis category.

**(1) Quality management policies:** Regarding this category of the analysis, three subcategories emerge: (1) strategies for preparation for accreditation, (2) meanings of the quality process and (3) benefits of quality management.

Among the main strategies used for the improvements implemented by the teams, reference is made to the gradual training processes, from the central level, which consider, in the first instance, a group of professionals who would take the lead to operationalize the quality policy in the CESFAMs. Subsequently, in the presence and support of the authorities of the municipal health departments, socialization spaces were created for all health center officials, where the main objective is to promote the purpose of incorporating quality processes in PHC.

“(…) As it was a big team, as I told you, I don’t know, we were 12–14, we were many, so we did training for the rest of the officials, as my colleague said, because they had to be enchanted and know what it was the quality, what was it for (…)”

These teams confront the different strategies and build their own meanings of the quality processes, identifying in the analyzed story different perspectives on the matter:

“(…) we saw quality, there was a lot of talk, but it seemed a bit far away, but later we realized that it was super important and that the quality process was coming now.”

Secondly, it is stated by the participants that the process was highly stressful and complex, that it should have been addressed with a sense of urgency, because there was a perception that the financing of the benefits provided in the centers would be an affected product of non-accreditation.

“Slow, difficult, for everyone, because the concept of quality did not exist, as one of the participants said, we hear it from far away, these are not the ones from Santiago, more or less, it will never happen to us. And when, well, I assumed the Explicit Health Guarantees (GES), and that it was important to accredit ourselves to be able to deliver GES benefits to, shoot this issue is important, and that’s when those of us who are directly affected began to take weight at first, in quotes those who we had scope.”

On the other hand, it is expressed that in the first years prior to the implementation of the structural conditions of the PHC that allow the centers to present themselves to the accreditation process, the teams dismissed the importance of quality, which is expressed in a context where the priority is accentuated in the fulfillment of the procedural clinical tasks, and the fulfillment of the health goals of primary health care.

“I remember that, in the old CESFAM, we did not give priority to quality … I don’t know if it really was like that … and here when we started seeing the importance … what it meant … no It may have been about 7 years, not if it was a long, long, long job.”

Finally, in relation to the main benefits perceived with respect to the quality process, the importance of having a reference system that guides the evaluation of the processes related to health care is identified, and that allows from that identification of needs to raise improvement plans that guarantee a higher quality of care and safety for the patient. This is content that is repeated in the different levels, the technical, administrative, professional and quality managers in the three macro areas interviewed.

Another achievement that the health teams perceive is being able to have an orderly record that allows them to identify their functions and support their responsibility, the latter as a protection strategy against legal conflicts.

Continuing with the benefits derived from accreditation, the participants of our study point out the following.

“(…) Well, in the CESFAMs, different processes are carried out, I don’t know, there is the issue of family health, there is the issue of quality, the issue of models and others that are being integrated. I feel that they are all models that allow you to see how things are done otherwise there is no capacity, you live day to day. And then, quality is like giving you guidelines to be able to look at, for example, user satisfaction. Evaluate how the rights of the patient are being respected, how certain procedures are being performed, adverse events. So, this is an instance that allows you to constantly see how you are doing things and seeing that possibility of improvement.”

**(2) Structure of primary health care:** Regarding this category, it was observed it had three subcategories: (1) administrative aspects, (2) financial aspects and (3) organizational aspects.

Regarding this category, the participants perceive the importance of formality and having an agenda but also the need for financing and its sustainability that guarantees quality processes over time.

“(…) I feel that the health service did the job well, in the sense that they began in the first instance educating the directors, the quality managers that at that moment one did not know what it meant and then later they kind of started to one trying to link the team and from there it was our decision in this case, to accredit us.”

In this case, for the teams to have a quality assurance strategy in health, it involves a formalized quality policy, a detailed action agenda and regulations that regulate its operation. Under this perspective, the administrators of the primary health system have the obligation to establish continuous improvement systems and programs that consider all the aspects involved.

The information collected made it possible to identify that the primary health administration facilitates the processes to the extent that training instances are generated at the different management levels, recognizing in turn the functions and times devoted to quality activities. This led to a significant effort in hiring personnel with functions specifically designed to lead the implementation of the quality and accreditation processes. In this way, the staffing gaps generated by the reconversion of the functions of the care team were covered, who were required by an increase in activities and additionally with a pandemic scenario, which further exacerbated this demand.

Regarding the financial aspects, the study highlights the lack of financing for accreditation, which is recognized by the participants. In this regard, it can be inferred that the concern of the teams also includes the sustainability of the process.

“No, I think … economic, limited resources. I believe that from now on we all have human resources destined for the activities we do daily. Of course, because we had to adjust to doing this, which is like an extra.”

The teams mainly adduced gaps associated with the structure that are related to financing, infrastructure and organization. In this regard, the teams state that the main difficulties and fears experienced are related to the conditions of the architecture of the health center, because its solution requires wills and resources external to the center, and on the other hand, the financing to carry out the accreditation process.

**(3) Participation and co-construction:** This category includes the (1) evaluation culture, (2) multilevel participation strategies and (3) characteristics of the quality team.

In this regard, the teams describe participation as a central element in accreditation, because it requires the involvement of all the actors in the care process.

“It took a lot for the health department to understand the importance of the person who was in charge of quality being exclusively for that and not sharing it with other areas.”

“(…) the Municipal Health Directorate (DISAMU), who are our direct bosses, are still empowered with what quality accreditation is (…)”

“I believe that the accompaniment of the authorities like the Regional Ministerial Secretariat (SEREMI), which is like ruling, how do they do it like this, they must have this, but there is no like a partner who supports you, we see it as an authority, a mandate. It is a mandate, so there is not a lot of flexibility with the establishments, we always talk about the fact that all realities are different, family health, etc., but here it does not apply much, so I see that from this point of view the SEREMI maintains a little … In other words, perhaps a companion would be that the processes in which we are failing … that could perhaps provide us with some type of help, a type of solution or something like that, because it is like, well, solve it yourself (…)”

Thus, the health teams value positively that the authorities are aware of the quality policy and support its implementation, not only with resources but also from their positions of power, generating spaces of trust to generate relevant action plans in local realities, to install accreditation processes in a timely manner.

On the other hand, the participation of users in this process emerges as an aspect to be strengthened, because their experiences are limited to instances of simulations. It is identified in the reports that the improvement plans are focused on the quality team who have responsibilities and greater training on the subject.

“(…) there was no such knowledge of this and when we began to investigate and we began to take courses and so on, we began to realize that in reality much of what is done in health care in an erroneous way depends on of the person who does it and not of the establishment, of the institution.”

“From the day we went to work, from the beginning of the day we knew that at any moment we were being evaluated by those teams that were supervising us, so to speak.”

“(…) we work in a network, we are no longer, I am not like me alone who thinks about everything, but we work in a network, and we are seeing the different realities that happen in all the centers.”

“In other words, if the service assistant participates in cleaning the unit, I have to call the service assistant to tell me, correct what needs to be corrected and see what she thinks of what I am proposing as well.”

“He let us be really, important, we have to push forward and quality every time there was something public, he said quality, accreditation.”

**(4) Leadership and change management:** Regarding this category, the skills of quality managers that allow influencing the practices of health teams to implement quality policies were recognized. Its main objective is to lead the processes toward safe and quality care. The continuous improvement process requires leaders capable of promoting and managing change in practices to improve the quality and safety of care. Therefore, it is described as the main facilitator for an optimal development of the processes and having minimum organizational conditions, such as time dedicated exclusively to the function of quality managers. In addition, the health team values positively that those in charge of quality have clinical or management experience in the same health center. This is because, according to the teams, they will have greater knowledge of the processes that have been installed over time and will have greater knowledge of the organizational culture. Communication skills and emotional intelligence stand out as facilitators of teamwork processes. On the other hand, when the person in charge of quality is new to the organization, an identified obstacle is not having quality training or experience in accreditation processes in other centers.

“(…) my main function is to lead this process, not only the accreditation process, but the idea is to lead the process of safe and quality care.”

“But the first time it was very difficult, for everyone … a few years ago … it was something like … he started with a person, who was the one who was given quality, and as colleagues say, since we didn’t catch it, we weren’t really interested … exactly, there was no interest … and then, from Little by little, DISAMU began to support in part, they were given more hours, then they were allowed to form a team (…)”

“I think that the quality manager would have been a colleague before, that he would have worked in the field and not that an external person would have been imposed on us. So the new manager knew the reality in which we were inserted, not that someone from outside came to impose, because she worked in a different way.”

“He had management experience. He had participated in accreditation processes, had a postgraduate degree in management, quality assurance. I had been in the accreditation process of the Atacama clinic, had worked and had seen what the accreditation process had been like at the Arturo López Pérez Foundation in Santiago (…)”

On the other hand, when the teams refer to the subsequent moment of accreditation, they report feeling satisfaction and pride, because having their health center accredited represents greater quality and safety for users. It is understood as a collective achievement, which makes it possible to reinforce the quality culture of the organization. Again, in this space, it is not possible to identify the consideration of the external user in obtaining the accreditation achievement.

For health teams, obtaining accreditation means satisfaction and development, a product of compliance with a series of new, complex and extensive processes. In addition, the culture of quality is strengthened because the health teams understand the importance of continuous improvement and can visualize the positive impact in their daily work.

Next, Figure 1 can be seen, which allows us to graph the facilitating and hindering aspects of the accreditation process, crossing them in turn with the 12 identified subcategories, which emerged from the four categories of analysis (the policies of the management of the quality, structure of primary health care, participation and co-construction, and leadership and change management). This allowed us to provide a condensed graph of what was previously described.

In this regard, it can be observed that the thickest shading corresponds to the weight or importance of each subcategory. In the first place, it is possible to observe a greater preponderance for the facilitating aspects versus the obstructing aspects.

On the other hand, for the facilitating aspects, the organizational aspects, the multilevel participation strategies, the leadership of the quality manager, the evaluation culture and the characteristics of the quality team are identified as the most relevant. Regarding the hindering aspects, the organizational factors stand out again, as well as the multilevel participation strategies, the financial aspects, the meanings of the quality process and the characteristics of the quality team. It is worth noting that the financial aspects and the meanings of the quality process had a greater relevance as hindering aspects.

## 4. Discussion

Quality accreditation is increasingly used in healthcare systems around the world. However, there is a lack of evidence which makes this study more relevant [26].

In this regard, as indicated in the results, four categories emerged from the study.

The first, related to “Quality management policy”, describes the strategies that allow preparing the CESFAMs teams in the implementation of quality management and the way to present themselves to the accreditation process. Faced with this, the health reform in 2005 involved the implementation of the system of the GES in Chile, incorporating the dimension of the quality in PHC [27]. However, despite the time that has elapsed, according to Forascepi Crespo [28], there are still serious problems in the management of the Chilean health system that conditions the existence of clinical services with deficiencies, and with gaps in the resolution capacity in PHC, with the negative consequences in the perception of the users [29]. In contrast, in a study in Chile in 2019 [30], they applied an instrument to evaluate the different dimensions related to the organizational structure of the CESFAMs, obtaining as a result a “good performance”, which is related to the fact that there is no unique mechanism to measure the quality of care.

According to the World Health Organization [31], “quality health care increases the probability of desired health outcomes and is consistent with the seven measurable characteristics: effectiveness, safety, people-centeredness, opportunity, equity, care integration and efficiency”.

Based on this, the teams recognize certain strategies, meanings and benefits that the accreditation process can bring them. All of this can somehow lead to improvements in the quality of care.

Regarding the above, a study carried out in Beirut in 2014 [32] establishes that the directors of 23 PHC centers affirmed that the accreditation led to an improvement in quality in several areas, the regression analysis showed that strategic quality planning, customer satisfaction and staff involvement were associated with higher perceived quality outcomes. Another study conducted in Qatar [33], a survey applied to 500 APS employees, agreed on the positive impact of accreditation. The results observed a positive and significant correlation between the staff’s perception of accreditation and the quality of care. The study goes on to state that workers perceived accreditation as a valuable tool for the quality of the organization. These results led to the conclusion that accreditation influenced the development of quality improvement practices and, therefore, had a positive impact on it. An improved compliance with accreditation requirements by healthcare organizations was shown to be a tangible indication of the effectiveness of organizations [33].

Despite the above, some discrepancies should be considered. In this regard, a study conducted in Egypt differs from the results of our study regarding the impact of accreditation, stating that accreditation had no effect on quality, which contradicts the positive findings [34].

In relation to the category “Structure of PHC”, it corresponds to the understanding of those elements that allow the administration, financing and organization of PHC and that influence facilitating or hindering the processes of quality and accreditation in the centers.

In this case, the study reveals the recognition of the importance of the administrative, financial and organizational aspects. Undoubtedly, the due assessment of the multiplicity of aspects considered in these management areas contributes to a greater commitment and empowerment of the teams involved [31].

The foregoing sheds some light, for example, on the importance of human resource management and strategies for the development and promotion of people’s competencies, for a better contribution to the quality management and accreditation process. In accordance with this, Saura Llamas [35] states that patient safety should be considered as an area of essential competencies, by itself and because it encompasses or is transversally related to many essential competencies already included, such as care communication, clinical reasoning, clinical management, teamwork, the management and organization of healthcare activity, information systems, healthcare quality, civil and medical-legal liability and bioethics. Therefore, through the development of this patient safety training strategy, many of these competencies could be acquired in a combined way, at least in part.

On the other hand, in terms of financial aspects, as we said, the importance of this aspect is highlighted as a key element of the process. In this regard, a review study of the available literature on PHC accreditation found that accreditation is not financed by most governments and is voluntary. However, in other countries, they offer financial incentives. This highlights why there is a paucity of research on the nature and acceptance of accreditation in the PHC sector [36].

On the other hand, the teams view with fear the gaps that they may observe in these areas of management, especially in those that depend on the will of the authorities, such as financing, because it can affect from the structure to the organization of the process. In this sense, Goyenechea [37] analyzed the gaps in the infrastructure of the health system in Chile, exposing the difficulties in having a construction system that considers the technical needs of health teams, users and health providers [38,39].

The teams also show a lack of certainty regarding the availability of financial resources for the improvement process, after accreditation. This is accompanied by a lack of follow-up in the post-accreditation stages by the [36] authority. Complementarily, a study that aimed to characterize the communes with accredited CESFAMs and that was recently carried out by the same research team concludes that the economic limitations in which the establishments are located are problems that have resulted in mainly those CESFAMs with a strong component of teamwork being those that have successfully obtained accreditation [19]. Regarding the category “Participation and co-construction”, it represents the participation strategies at the different levels of primary care management, reflecting central elements of the evaluation culture, which have a direct link with the characteristics of the quality team. In this regard, the Superintendence of Health points out that in the first stage of the planning and implementation of quality, managerial commitment, the empowerment of leaders and quality training are essential to involve all the personnel of the institution [40]. For this reason, the study highlights this aspect, given that, in this sense, the participation of authorities or key personnel in positions of power and health decision making play an important role when it comes to listening, facilitating understanding between different interest groups (e.g., management teams, healthcare workers and healthcare consumers) and navigating bureaucratic structures to influence resource allocation and outcomes [41].

Finally, regarding the category “Leadership and change management”, the participants perceive leadership and the characteristics or conditions in which it is put into practice at the service of the accreditation process. Faced with this, there are numerous publications that recognize leadership style as a key element for the quality of health care. Effective leadership is among the most critical components that drive an organization to effective and successful results. Significant positive associations were reported between effective leadership styles and high levels of patient satisfaction and reduced adverse effects [42].

According to the above, the transformational leadership style is characterized by creating relationships and motivation among staff members. Transformational leaders often have the ability to inspire trust and respect from staff and communicate loyalty through a shared vision, resulting in higher productivity and building morale [43].

The above, that is, everything that is related to the leadership style and the characteristics displayed by the leader, can contribute to a more participatory management. In this regard, Doricci et al. [44] highlights the value of co-construction in PHC with the participation of all, in addition to a leader who practices horizontal communication, team mediation and assertive conflict resolution.

Derived from the above, the meanings of the process also emerge. In this regard, according to 11 surveyed PHC centers in Denmark, working with accreditation standards had a positive impact that was less tangible than specific changes in behavior and physical infrastructure. Some believed that the accreditation process had improved their abilities to engage in future quality improvement activities by making them more aware of where to look for relevant information, more aware of how to structure quality discussions in the clinic or better at comparing their own practice with official quality standards. Finally, several mentioned that the documents describing the workflows and the division of tasks (written during the accreditation process) could be seen as a new knowledge resource that could be used to facilitate the incorporation of new personnel in their institutions [45].

## 5. Limitations

Some of the limitations observed in this study derive from the fact that this is a qualitative analysis; therefore, the results cannot be generalized because the sample size is small and represents 21.4% of the total of the PHC centers accredited at the date of the study.

On the other hand, it is feasible to consider some selection biases in the sample. Even though all the accredited PHC centers were included, there are some that could not be incorporated into the analysis sample, mainly due to self-exclusion and economic barriers.

Lastly, it is possible that the study was exposed to a response bias; however, an attempt was made to control this with the questions of the semi-structured interview, which included different aspects of the accreditation process (facilitators and/or obstacles). In general, the staff could have responded by pointing mainly to the positive aspects of the process, given the final success achieved, which in general tends to have an impact over any negative memories that may have arisen.

## 6. Conclusions

The health teams participating in the study agreed that the accreditation of the CESFAMs can contribute to the improvement in primary health care, which benefits both internal and external users, laying crucial foundations within the health system.

On the other hand, the teams express the need to have minimum conditions for compliance with the accreditation process, such as the systematic training of the teams, especially those who lead the process.

Consistent with this, importance is given to the characteristics and conditions of leadership, because the presence or absence of certain technical and transversal competences can result in facilitating or hindering factors in the accreditation process, where harmonizing teamwork is an issue, a key factor to progress from the individual perspective to the vision of the entire organization.

Another point that the teams show is that historically care work has had to be favored over management activities due to limited resources. However, after obtaining the quality accreditation certification, its importance and benefits are recognized, thanks to the permanent support of the health authorities and the commitment of all those involved. Therefore, the sustainability of the accreditation process is important, which implies guaranteeing additional and permanent resources.

Finally, the participation of all those involved in the health care process is described as a central element in quality and accreditation processes, highlighting that, in the future, the active participation of external users (community) should also be included. For now, this role has been only receptive. Therefore, the idea arises to continue in this line of research, with projects that analyze and promote instances of more active participation of the community in this important process.

Regarding the practical implications of this study, the first thing is that it contributes to reducing the knowledge gap on accreditation processes, including their results and effects both on health teams and on the external user (patients/community).

Secondly, the study can be an important input and support for non-accredited CESFAM teams, to encourage them to undertake their own paths toward achieving quality accreditation certification.

Finally, one last implication is that this material becomes a relevant input for those in charge of formulating public policies that strengthen primary health care.

## Figures and Tables

**Figure 1 ijerph-20-02477-f001:**
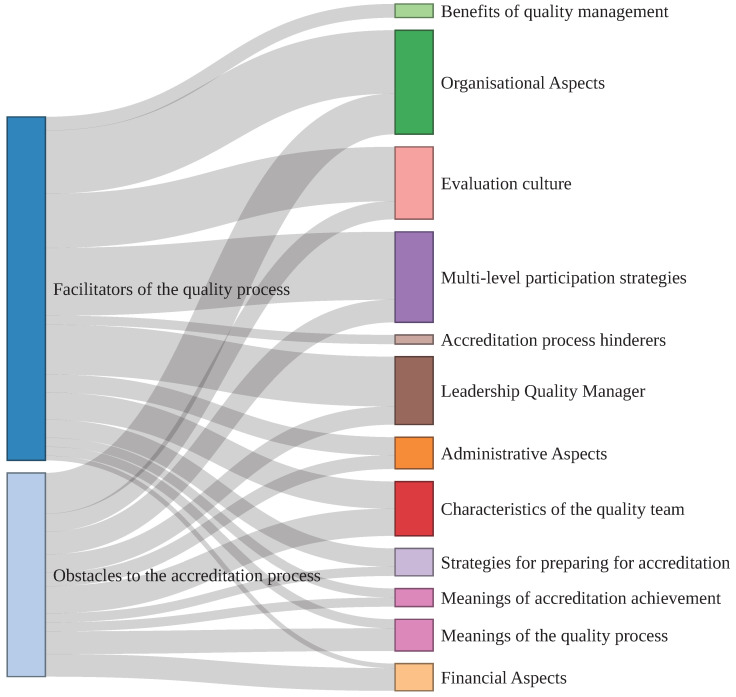
Graphical representation of the facilitating and hindering aspects.

**Table 1 ijerph-20-02477-t001:** Characteristics of Quality Managers.

N°	Profession	Age	Gender	Time in Office	CESFAM	Commune	Region
1	Psychologist	33	Male	4 years	Candelaria del Rosario	Copiapó	Atacama
2	Kindergarten educator	39	Female	5 años	Juan Martínez	Copiapó	Atacama
3	Nurse	38	Female	4 years	Paipote	Copiapó	Atacama
4	Kinesiologist	35	Female	4 years	Dra. Michelle Bachelet Jeria	Chillán Viejo	Ñuble
5	Midwife	38	Female	7 months	Isabel Riquelme	Chillán	Ñuble
6	Kinesiologist	35	Female	7 years	Los Volcanes	Chillán	Ñuble
7	Nurse	31	Female	3 years	Sol de Oriente	Chillán	Ñuble
8	Kinesiologist	35	Female	5 years	Ultraestación Dr. Raúl San Martín	Chillán	Ñuble
9	Nurse-Midwife	65	Female	10 years	Dr. Jorge Sabat	Valdivia	Los Ríos

**Table 2 ijerph-20-02477-t002:** Categories and Subcategories.

N°	Core Categories	Subcategories
1	Quality management policies	Accreditation preparation strategies
		Meanings of the quality process
		Benefits of quality management
2	Structure of PHC	Administrative aspects
		Financial aspects
		Organizational aspects
3	Participation and co-construction	Culture of evaluation
		Multilevel participation strategies
		Characteristics of quality team
4	Leadership and change management	Leadership from people in charge of quality
		Meanings of accreditation achievement

## Data Availability

The data are not publicly available due to the protection of the privacy and identity of those who gave informed consented to participate in this study. Said data protection was approved by the Institutional Ethics Committee: Comité de Ética Institucional Universidad de Santiago de Chile.

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
