# Peer review of "Accreditation of Quality in Primary Health Care in Chile: Perception of the Teams from Accredited Family Healthcare Centers"

_ijerph, 2023, doi:10.3390/ijerph20032477_

Round 1

Reviewer 1 Report

The paper is well organized and with the objectives clearly set out and an appropriate discussion of the methodology is opted. The collection and the use of the data is creditable and in the end the results and discussion are quite impressive. However, I found few minor Typo / formatting slip-ups as mentioned bellow.

References require minor revision as two different format styles are used to write down the name of the studies. For example, see lines 720, where Capitalize Each Word format style is used as opposed to Sentence case format, which is used for most of the references. Further, the same issue has been observed while writing the journal names. The author(s) are required to fix such issues for the sake of uniformity. Further, few incomplete references are also noticed.

Reviewer 2 Report

The study topic is interesting.

However, this manuscript requires extensive revisions:

1. The manuscript should be checked by a native English speaker.

2. The manuscript should be rewritten - please avoid "we", and replace it with "this study" etc. This is a scientific paper, so please follow the scientific wording. This refers to both the abstract and main text.

3. Please provide more international background in the Introduction section. Please justify, why this study is important for international readers.

4. Please clearly define the aim of the study. This study aimed to.....

5. Please revise the methods section to provide clearly define study procedures. The Authors may divide the methods section into sub-sections.

6. In Table 3, the title is in Spanish.

7. The results section is unclear. Please provide well-structured and logical text. The Authors may move some parts to the supplement or focus only on the most important findings.

8. Please provide 2-3 sentences on the practical implications of this study and further research needs.

9. Conclusions are too long. Please provide more informative and comprehensive conclusions.

Round 2

Reviewer 2 Report

The manuscript was revised in line with the suggestions provided by the reviewer.